

# The Sub-Polar Gyre Index - a community data set for application in fisheries and environment research

Barbara Berx[1] and Mark R. Payne[2]

[1]Marine Scotland Science, 375 Victoria Road, Aberdeen, AB11 9DB, UK
[2]Centre for Ocean Life, Technical University of Denmark, National Institute of Aquatic Resources, 2920 Charlottenlund, Denmark

*Correspondence to:* Barbara Berx (b.berx@marlab.ac.uk)

**Abstract.** Scientific interest in the sub-polar gyre of the North Atlantic Ocean has increased in recent years. The sub-polar gyre has contracted and weakened, and changes in circulation pathways have been linked to changes in marine ecosystem productivity. To aid fisheries and environmental scientists, we here present a time series of the Sub-Polar Gyre Index (SPG-I) based on monthly mean

maps of sea surface height. The established definition of the SPG-I is applied, and the first EOF and PC are presented. Sensitivity to the spatial domain and time series length are explored, but found not to be important factors. Our time series compares well with indices presented previously. The SPG-I time series is freely available online (http://dx.doi.org/10.7489/1806-1) and we invite the community to access, apply and publish studies using this index time series.

## 1 Introduction

The sub-tropical and sub-polar gyres are the dominant features of the surface circulation of the North Atlantic Ocean (Figure 1). Both are driven by the combination of permanent wind features, heat-input variation with latitude and the global overturning circulation. The sub-tropical gyre is formed by the synthesis of the Gulf Stream, North Atlantic Current, Canary Current and North Equato-

15 rial Current to yield a nearly continuous, anticyclonic circulation in the sub-tropical North Atlantic Ocean. Its equivalent in the sub-polar region can be considered as a cyclonic gyre encompassing the North Atlantic, East Greenland and Labrador Currents (Figure 1). Within the North Atlantic, changes in the strength and extent of the sub-polar gyre have been linked to changes in the advection of water masses (Häkkinen and Rhines, 2009), and changes of their properties (Holliday et al., 2008;

Johnson et al., 2013). These changes in the strength and extent of the sub-polar gyre have been attributed to the strong overturning circulation observed in the preceding years (Häkkinen and Rhines,



2004; Bersch et al., 2007). More recently, marine ecologists have reported changes in the ecosystem associated with changes in circulation in the North Atlantic, and particularly the sub-polar gyre region (e.g. Hátún et al., 2009b; Hátún et al., 2009a; Payne et al., 2012).

Based on a survey of research scientists within the International Council on the Exploitation of the Seas (ICES) community, the Working Group on Operational Oceanographic products for Fisheries and Environment (Berx et al., 2011, WGOOFE;) identified a need from fisheries and environmental scientists for freely accessible oceanographic data, in a suitable data format and with operational delivery. Within the climate community, the need for large volumes of data to be distilled into readily accessible, user-friendly data sets has recently driven the development of the Climate Data Guide (Schneider et al., 2013; Overpeck et al., 2011). This site provides a community-based overview of available data products, and includes some expert guidance on the strengths and weaknesses of the products as well as information on their derivation. In its work as an interface between the ICES community and the operational oceanography community, WGOOFE found that index-based products —where a complex spatio-temporal data set, process or state estimate may be reduced to a single time series, such as the North Atlantic Oscillation Index (NAO; Hurrell & NCAR Staff, 2013)— remain a major gap in the available oceanographic data products (ICES, 2012a). Recently, Bessières et al. (2013) presented three index-based oceanographic data products developed by the MyOcean project: the El Niño indicator, the Kuroshio Extension indicator and the Ionian Surface Circulation indicator. However, these index time series have not yet been made readily available to the wider community. To date, no readily accessible up-to-date dataset exists summarising the sub-polar gyre's dynamics. This limits some researchers in the field of ecosystem science in the search for drivers of ecosystem variability within the region, and therefore also the development of improved fisheries management tools. Furthermore, the ICES Working Group on Widely Distributed Stocks (WGWIDE) have highlighted the absence of such a data product as a key obstacle when studying economically important fish stocks, such as mackerel and blue whiting (ICES, 2012b).

The dataset presented here aims to fill this gap in index-based operational oceanographic data products by presenting a time series of the Sub-Polar Gyre Index (SPG-I) extending from the start of satellite altimetry records (January 1993) to the present. The dataset presented is freely available, and easily citable. In Section 2, we outline the underlying dataset and methodology for our SPG-I calculation; in Section 3 we present the data product and its sensitivities, and compare our timeseries with other published results of SPG-I variability; and finally we present how to access the SPG-I data product and acknowledge its use (Section 4), followed by a brief conclusion and outlook (Section 5).



## 2 Methodology

### 2.1 Sea Surface Height Data

The altimeter products used to create the SPG-I were obtained through the Copernicus Marine Environment Monitoring Service (CMEMS; product identifier: SEALEVEL_GLO_SLA_MAP_L4_REP _OBSERVATIONS_008_027). For our analysis, we obtained the delayed time, global, daily Maps of Sea Level Anomaly (MSLA) on a $1/4^o$ by $1/4^o$ grid. The product is the result of merging all available satellite missions at a given time, resulting in a better quality product (particularly in recent years). Monthly mean maps were created by averaging the multimission daily maps by month, while seasonal climatology maps were calculated by averaging the monthly mean maps within the same month for all complete years (1993-2015). The climatological maps therefore represent the average conditions for each month of the year throughout the observation period. To avoid issues with observations in grid points on land, a land-sea mask was obtained by interpolating the $1/12^o$ by $1/12^o$ TerrainBase database (National Geophysical Data Center, 1995) on to the same grid as the altimeter data. The land-sea mask was also used to remove altimeter data in the Pacific Ocean, Great Lakes and Mediterranean Sea. The altimeter data set starts in January 1993, and the latest update obtained from CMEMS extends to May 2015.

### 2.2 Calculation of SPG-I

We followed the method of Häkkinen and Rhines (2004) to calculate the SPG-I, which has been defined as the first Principal Component (PC1) of an Empirical Orthogonal Function Analysis (EOFA) of the sea level anomaly field in the North Atlantic. In Section 3, our results are compared to similarly defined gyre indices based on altimeter data, although alternative indices based on sea surface temperature and wind stress curl have also been defined by Hátún et al. (2009a) and Häkkinen et al. (2011), respectively.

In our analysis, we restricted the geographical extent to a rectangular area focused on the North Atlantic's sub-polar gyre (delimited by the $60^o$ W and $10^o$ E meridians, and the $40^o$ N and $65^o$ N parallels). The exact choice of spatial domain varies between authors - here we have chosen a region focused on the sub-polar gyre region itself. However, we also perform sensitivity analyses to examine the effect of this choice on the resulting index time-series. Seasonality in the monthly mean observations of sea level anomaly was removed by subtracting the relevant climatological sea level anomaly map. We calculated the SPG-I based on these deseasonalised maps of sea level anomaly.

EOFA is a well-established analysis technique, but for completeness a short description follows. For more in depth information, we refer the reader to Emery and Thomson (2001).

A major strength of EOFA is the reduction in data volume: a large dataset can be reduced to a smaller one containing still the most significant fraction of variability contained in the original data. In particular, it can often reduce large spatial data sets to a more manageable size. We can consider





the altimeter data set as a time series of $I$ time instances at $J$ locations, defined by the point's
latitude and longitude on the grid. During EOFA, the data matrix is standardised (for each location,
the mean is removed from the time series and the remaining anomalies then scaled by dividing
by their standard deviation) and then decomposed into mutually uncorrelated (orthogonal) modes
which have a spatial pattern (these are called the eigenvectors or empirical orthogonal functions)

and a temporal amplitude (these are called the eigenfunctions or principal components). The first
mode extracted using the EOFA technique explains the largest fraction of variability in the data set,
and each subsequent mode explains the largest fraction of the remaining variability. By extracting
the first mode, we obtain the time series of SPG-I based on the MSLA data. The sign of an EOF/PC
is not determined explicitly during the calculation process, and may vary between machines and

software versions. We therefore define the sign of the EOF and PC in 1993 to be positive, and
make adjustments as required. To highlight inter-annual variability, the SPG-I time series has been
smoothed using a 13-month running mean.

The addition of data points either in space or time also changes the EOF/PC results, which is
briefly explored here. For the user, this means the entire time series needs to be updated when the

temporal coverage of the data is updated. To investigate the impact of the chosen spatial extent of
altimeter data used in the EOFA, we performed the analysis on a number of different regions. These
were chosen to cover a number of regions centred on the sub-polar North Atlantic. These results
were based on the $1/4^o$ by $1/4^o$ grid, although time series based on $1^o$ by $1^o$ grid and $2^o$ by $2^o$ grids
are also presented. The longitude/latitude limits of these regions are listed in Table 1 and shown in

Figure 3.

### 3    SPG-I time series

Figure 2 shows the first Principal Component (i.e. temporal variability) of the SPG-I and correspond-
ing Empirical Orthogonal Function (i.e. the spatial field). The spatial field shows the signature of the
Gulf Stream/North Atlantic Current variability (eddy-like features in Figure 2a), as well as the lower

sea level in the sub-polar gyre region compared to the sub-tropical Atlantic. This first mode of the
EOFA explains 26.2% of the total variance. In the SPG-I time series (Figure 2b), positive values of
the index are associated with a strong sub-polar gyre circulation with a wide spread. In comparison,
negative values of SPG-I are associated with a weak sub-polar gyre and westward retraction. The
commonly reported weakening and contraction of the sub-polar gyre can be seen in the mid to late

nineties. More recently the sub-polar gyre has been variable but remains weak (Figure 2b).

### 3.1    Sensitivity to spatial extent

To investigate the sensitivity of the SPG-I to the chosen spatial domain for the EOFA, the analysis
was repeated for a number of different regions (Figure 3 and Table 1). The results, shown in Fig-



ure 4a, show a separation does occur between indices focused solely on the sub-polar region (time
series B, 6, 7 and 8 in Figure 4a), and those incorporating a wider region of the North Atlantic.
In particular, when including the region in the tropical North Atlantic Ocean, the SPG-I becomes
strongly linear. The correlations between time-series based on these various regions are high (all
greater or equal to 0.92, p<0.001). The correlation coefficients also highlight that indices calculated
from regions restricted to the sub-polar North Altantic correlate well to each other but less-well to
those calculated over the wider North Atlantic, with r values dropping from 0.99 to approx. 0.93.
The SPG-I calculated for regions narrowly confined to the SPG-I North Atlantic have a higher level
of inter-annual variability.

## 3.2 Sensitivity to time series length

To highlight the impact of increasing time series length on the SPG-I, the analysis was performed
on the first 10 years of altimeter data (1993 to 2012), in time increments of one year (Figure 5).The
analysis highlights the need to access the latest SPG-I time series, in order to use the most accurate
representation of sub-polar dynamics in further analyses. The convergence of the index time series
seen in Figure 5a suggests that the time series is now of sufficient length, and the definition of SPG-I
as the first PC of an EOFA of SLA is robust.

## 3.3 Comparison to previous results

The SPG-I presented here has been compared to similar indices previously presented by S. Häkkinen
(Häkkinen and Rhines, 2004, 2009) and H. Hátún (Hátún et al., 2009b, a). A first comparison (Figure
6) shows the comparison of the monthly resolution SPG-I time series to that previously presented by
S. Häkkinen (S. Häkkinen, pers. comm.). These two time series show close agreement, and minor
differences are likely due to different underlying altimeter data products and the change in time series
extent. There is also a difference in scaling between these two indices by two orders of magnitude,
most likely due to differing units (centimetres vs. metres). However, as the typical useage of the
data set is in terms of its correlations with other variables, rather than interpreting its absolute value,
this discrepancy is not seen as important. In a second comparison (Figure 7), the annual filtered
time series of both these indices has been compared to the time series estimated by H. Hátún based
on annual mean SSH data (H. Hátún, pers. comm.). Again, the overall pattern of these three indices
shows good agreement, with all showing a clear reduction in the SPG-I in the mid to late nineties. The
difference between this third index time series in comparison to the previous two is due to a different
underlying altimeter product (a $1/4^o$ by $1/4^o$ Mercator grid which placed additional emphasis on the
sub-polar gyre region).



## 4 Data Access

Following the recommendations of Berx et al. (2011), we would like to ensure the SPG-I time series presented here is easily accessible and available to all. The data can therefore be downloaded from http://dx.doi.org/10.7489/1806-1 in ASCII format, and researchers can also access the supporting information there. The time series will be updated within 6 months from the time the data provider (CMEMS) publishes its updates. As the time series is based on the EOFA technique, we recommend users to download the entire index time series when updating the time series they use. This will ensure the most relevant time series is used in any analyses. Version numbering will facilitate users identifying which version of the time series they are accessing. Finally, we would appreciate all those making use of this time series to appropriately acknowledge its use by citing this paper and the Digital Object Identifier (DOI) of the dataset.

## 5 Summary

We have presented a time series of SPG-I based on monthly mean SLA maps obtained from CMEMS. Our time series compares well with indices presented previously. The variability in the index time series is influenced by the chosen spatial extent of the sea surface height data included in the EOFA, and inter-annual variability is suppressed when including the wider North Atlantic region. We also presented an indication of changes with temporal coverage of the data series, and users need to ensure downloading the entire time series when accessing future updates. The index data product we present is freely available from http://dx.doi.org/10.7489/1806-1, and we encourage all scientists interested in establishing linkages between North Atlantic climate variability and ecosystem function to access, apply and publish.

*Acknowledgements.* We would like to thank Sirpa Häkkinen and Hjalmar Hátún for providing their time series of SPG-I, as well as ancillary information relating to their calculation; and to Hjalmar Hátún and Stuart Cunningham for useful discussions on sub-polar gyre dynamics. The research leading to these results has received funding from the European Union 7th Framework Programme (FP7 2007-2013) under grant agreement number 308299 (NACLIM).



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



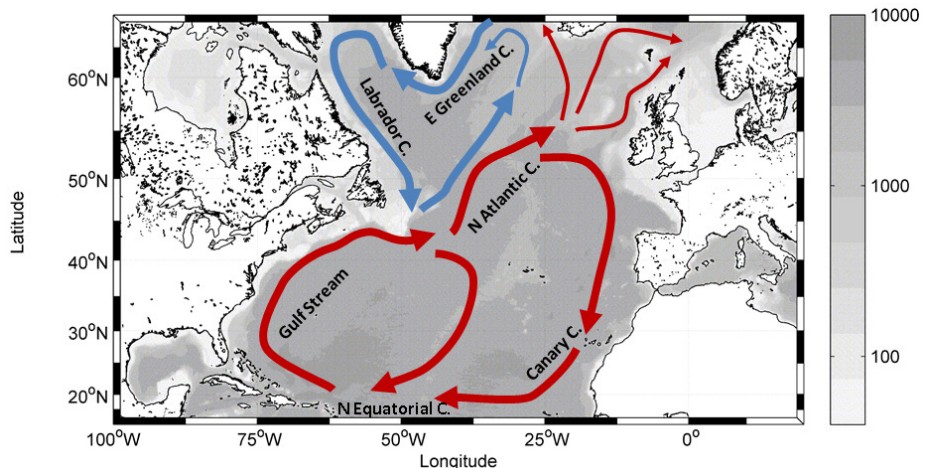

**Figure 1.** Map of the subtropical and subpolar North Atlantic with a schematic representation of the ocean circulation.

| Region | Longitude | | Latitude | |
|---|---|---|---|---|
| | Left | Right | Bottom | Top |
| S | $100^o$ W | $20^o$ E | $15^o$ N | $65^o$ N |
| B, 1X1, 2X2 | $60^o$ W | $10^o$ E | $40^o$ N | $65^o$ N |
| R1 | $100^o$ W | $30^o$ E | $0^o$ N | $66^o$ N |
| R2 | $95^o$ W | $25^o$ E | $5^o$ N | $66^o$ N |
| R3 | $90^o$ W | $20^o$ E | $10^o$ N | $66^o$ N |
| R4 | $85^o$ W | $15^o$ E | $15^o$ N | $60^o$ N |
| R5 | $80^o$ W | $10^o$ E | $15^o$ N | $60^o$ N |
| R6 | $75^o$ W | $5^o$ E | $30^o$ N | $66^o$ N |
| R7 | $65^o$ W | $0^o$ E | $45^o$ N | $66^o$ N |
| R8 | $60^o$ W | $5^o$ W | $45^o$ N | $66^o$ N |

**Table 1.** Overview of defined regions to investigate sensitivity of SPG-I to the chosen spatial coverage (regions are also shown in Figure 3). The abbreviations in the first column correspond to those used in figure legends. S = S. Häkkinen time series; B = time series presented here; R = time series based on different regions; 1X1 = time series presented here based on a $1^o$ by $1^o$; 2X2 = time series presented here based on a $2^o$ by $2^o$, all other time series presented here based on a 1/4$^o$ by 1/4$^o$.





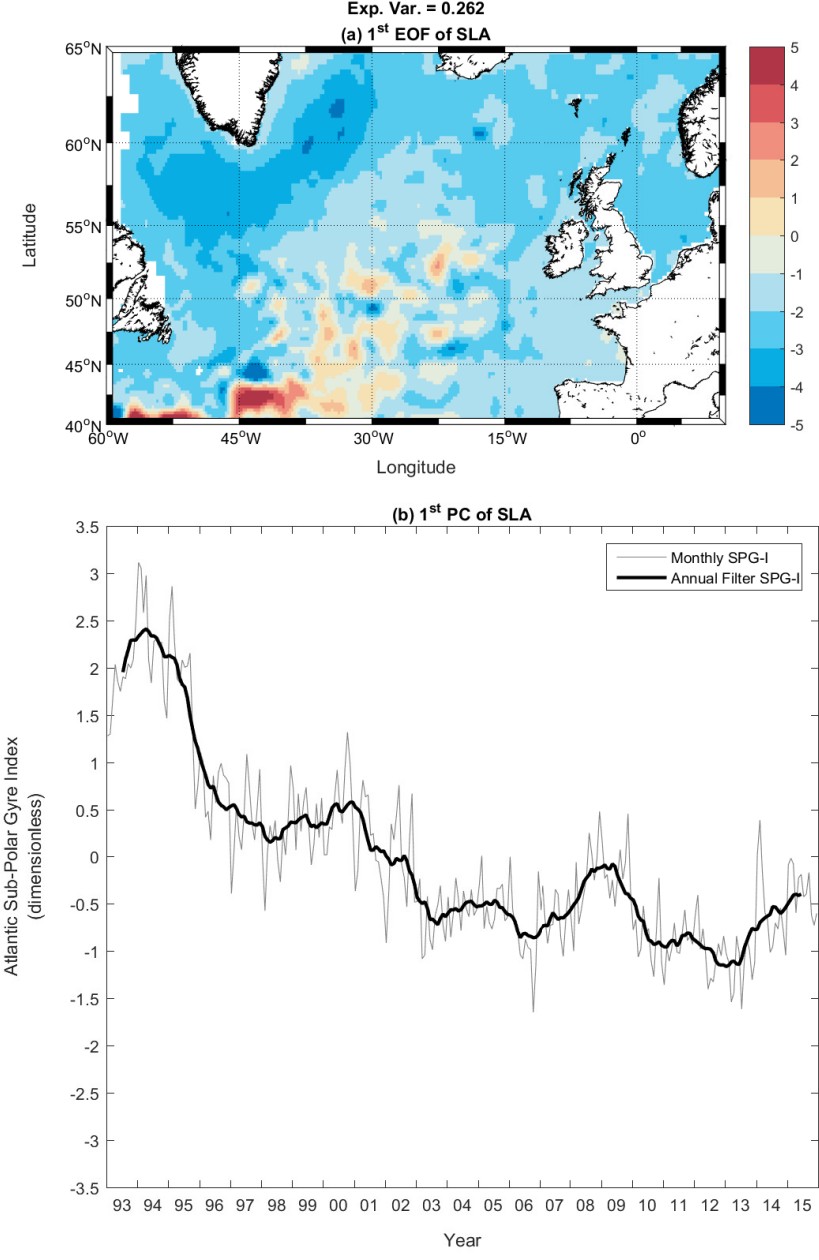

**Figure 2.** First Mode of the EOFA of sea surface height: (a) Empirical Orthogonal Function (spatial field, in cm); (b) Principal Component (temporal variability, dimensionless).




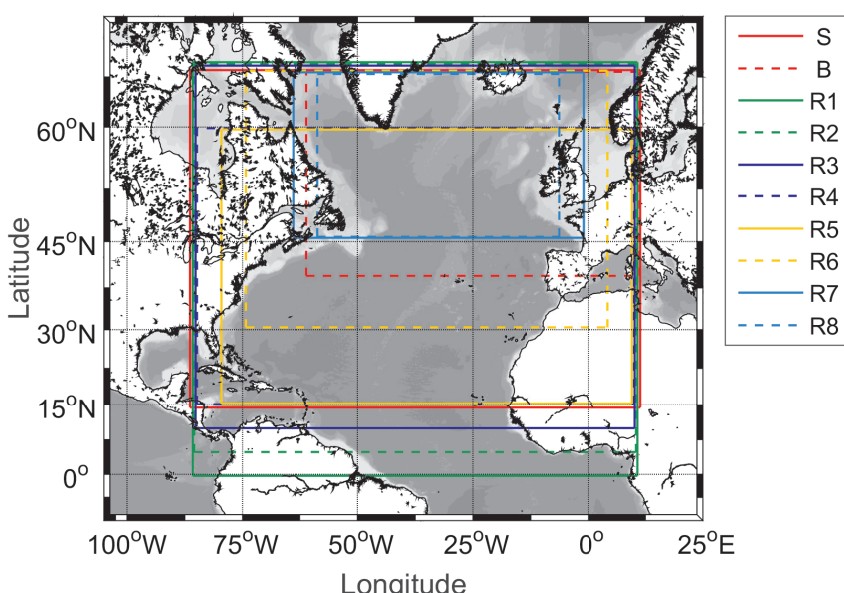

**Figure 3.** Map of defined regions to investigate sensitivity of SPG-I to the chosen spatial coverage. Details of boundaries are in Table 1. S = S. Häkkinen time series; B = time series presented here; R = time series based on different regions; 1X1 = time series presented here based on a $1^o$ by $1^o$; 2X2 = time series presented here based on a $2^o$ by $2^o$, all other time series presented here based on a $1/4^o$ by $1/4^o$ grid.





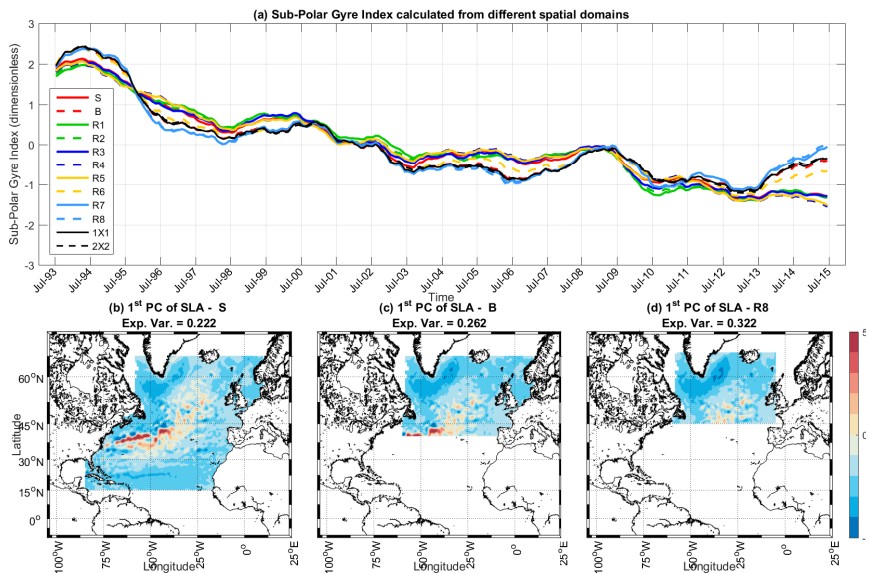

**Figure 4.** Comparison of annual smoothed SPG-I for different domains within the North Atlantic: (a) annual smoothed SPG-I time series, (b-d) corresponding Empirical Orthogonal Function (spatial field, in cm) for three domains. Table 1 and Figure 3 show the defined areas used in the EOFA. Colours in (a) correspond to those used in Figure 3



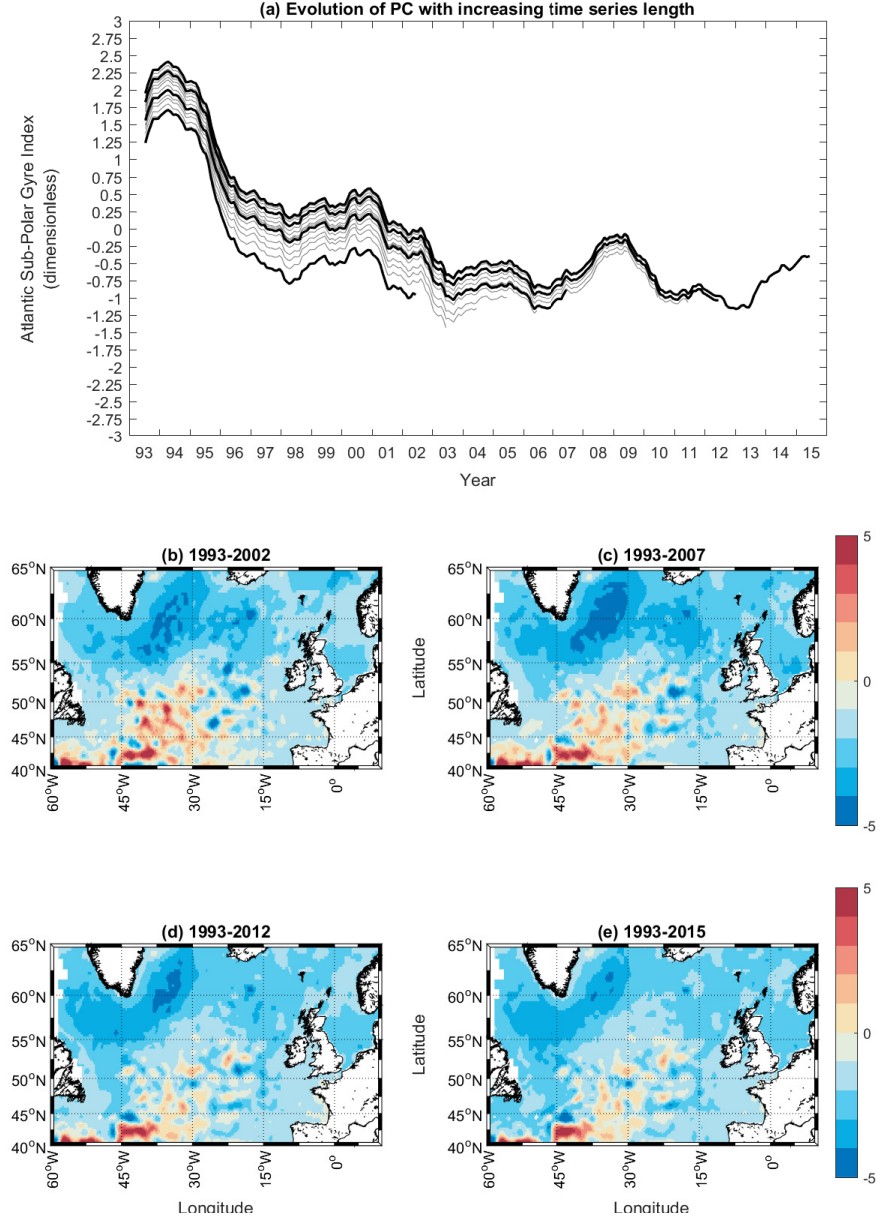

**Figure 5.** Sensitivity of SPG-I to time series length: (a) Principal Component (temporal variability, dimensionless) for time periods from 1993 to 2002, increasing one year in length (grey lines) with periods shown in (b-d) in bold black lines; Empirical Orthogonal Function (spatial field, in cm) for four time periods: (b) 1993 to 2002, (c) 1993 to 2007, (d) 1993 to 2012, and (e) 1993 to 2015.





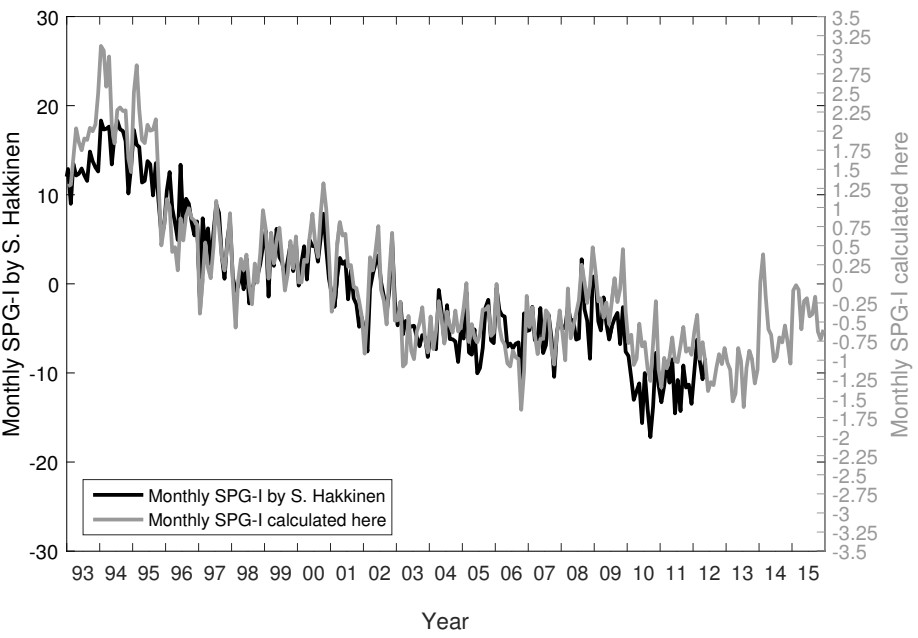

**Figure 6.** Comparison of SPG-I with that calculated by S. Häkkinen, based on monthly time series

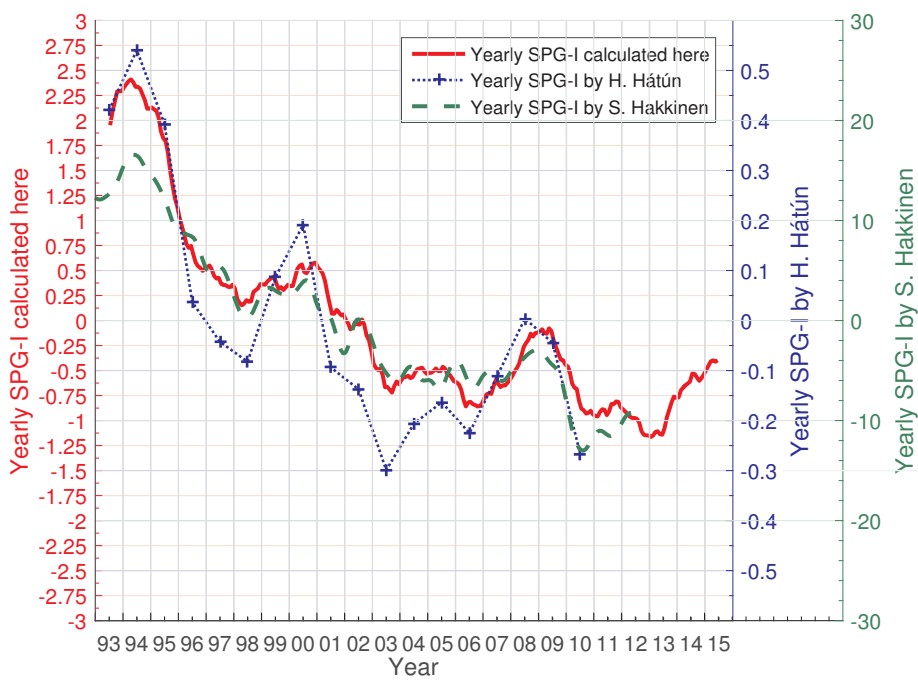

**Figure 7.** Comparison of annual smoothed SPG-I with that calculated independently by S. Häkkinen & H. Hátún. See text for more details.