# Peer review of "The Sub-Polar Gyre Index - a community data set for application in fisheries and environment research"

_Earth System Science Data, 2016_

## Referee Comment (RC1) · Anonymous Referee #1 · 15 Dec 2016

The authors present a new dataset for general release that provides an index of North Atlantic subpolar gyre circulation. The index has been presented in journal articles in the past, but never freely distributed or updated before now. The index itself should prove very useful for applied research and monitoring, and for those who are not able to re-create the index from the original data. The manuscript gives a clear description of how the index is computed, assessing it's sensitivity to the area over which is calculated, and the effects of the lengthening time series.

The manuscript is clearly written and provides good and appropriate figures. Overall my view is that it is suitable for publication with some revisions as follows.

In Section 3 you state that the index is EOF1 and explains 26.2% of the variance.

It would be useful here to have a fuller explanation of what the EOF is in terms of the physical changes in the SPG (and cite some papers that are the origins of the information). For example, what does "weak" mean in this context? Can you plot composites or examples of what the SSH anomaly looks like under negative/weak conditions vs positive or "strong: conditions? Or SST to show the "westward retraction" (what does this retraction refer to?).

It would be interesting to know how much of the variance is explained by the second/third mode and what those patterns might represent? Could they be useful indicators of a different aspect of the SPG characteristics?

On Line 125 you cite some numbers referring to Figure 4a, but I cant see what they refer to. In the same paragraph you discuss the correlations between the indices over several different regions. Do the correlations decrease if you use de-trended series? This might highlight the differences between the subpolar and wider regions.

I wasn't very convinced by section 3.2, the sensitivity to length of time series. I think you are saying that you need a decent length of series to get a robust result. The statement on line 136 suggests that users need always to access the latest version of the index to use the "most accurate representation of sub polar dynamics", but then go on to say that we have now reached a series of sufficient length (which suggests that at some point that first statement stopped being true). Anyway it looks to me as those the length of the series is not a problem even for the shorter series; if those indices in Fig 5 were all overlain instead of being slightly offset at the start, their apparent difference would look even smaller. And since the value in the index is not it's absolute value, but relative periods of high or low (as you say in the next paragraph), I think I would summarise this section as showing the form of the index is not at all affected by the number of years of data used.

In section 3.3 I noticed that you refer to your new index in Fig 7 as "yearly" data when there is clearly data at sub-annual time scales included in the red curve. In the text

you described the data as annually filtered - you just need to be consistent with the terminology in Figure 7, or ideally, would present actual yearly values from your new index.

---

## Referee Comment (RC2) · Anonymous Referee #2 · 16 Dec 2016

An index that has been used in other papers is discussed and a new update is prvided with an analysis of the sensitivity of its estimations, based on published/publically available SSH mapped products.

This is probably a valubale effort and this index will be used by some. However, I would like to see major improvements in the paper for the folling reasons:

Among major reasons provided for this index work is a recommandation from a working group (WGOOFE) of ICES. This does not seem enough motivation, or at least this is not argued enough.

How will this particular index help the Fisheries work? and how is it complementary to

other indices that are published in delayed mode (based for example on atmospheric variables (NAO, EA, Arctic Oscillation...) or on CPR survey data...). Or, to take an other example, on the time series and ocean analysis published by an other ICE working group (WGOH)?

On the other hand, one can guess that this index might be interesting for fisheries work, as this community may not be used to manipulating large gridded datasets (such as the ones provided by Copernicus Marine Service, for example T-S 3D analyses either from data (for example ISAS or ARMOR products) or model simulations).

I am also wondering about the interest to provide an index in a rather delayed mode (my understanding of the paper is that the 13-month smoothed index that is recommanded ends in May 2015; on the web-site, the monthly non-smoothed files end up in December 2015, with the next 6-month release from Aviso just published this week). Is there a commitment of the Scottish Institute to update the index? Or should it be defered once publication done to the Copernicus Marine Service (there is index work planed to be provided, but I dont think that it includes this index?); There also near-real time altimetric products that might be used to extend the time series to near-present (but this requires more work).

Mabye in the itroduction, it could also be informative to add bibliography, for example from modeling work on this index such as in the Gao Yong-Qi and Yuh Lei 2008 's paper or on variability in the subpolar gyre and connection to subtropical gyre?

I am also wondering whether this first EOF of SSH is the only part of the SSH mapped data that might be interesting to the fisheries community (or climate community). If this is research in progress, it could be worth adding indices of intergyre transport and different gyre intensity... Combining with other indices that can be derived from easily accessible indices could als obe helpful, but this requires more work (it could be an average T-S or density 0-1000m of the subpolar gyre?)

Then, different sensitivity tests are presented to show how stable and reliable is the

index chosen. Work is presented on tests of size of domain and smoothing. I dont see tests on whether normalizing the variance in each (spatial) grid point before EOFA could have an effect, whether. ANother point to test would be whether removing a time series of spatial mean before doing the EOF/PC analysis has an impact. There are also tests on time series length, butthey all include the first part of the time series, where the largest changes occur in PC1. This is not very informative, and more sophisticated tests to provide information on the stability of the pattern. This could be done by extracting EOF1 by randomly selecting subsets of years, and providing tests of significance (how, does the proportion of variance explained by EOF/PC1 changes, whether patterns and regressed time series vary or not...). Even, doing the analysis separately on first and second halves of the record could be instructive (instead of figure 5)

Minor comment. The schematic circulation of Figure 1could be modified/updated. Not that great from a physical point of view.

---

## Referee Comment (RC3) · Anonymous Referee #3 · 2 Jan 2017

In my opinion, the manuscript is suitable for ESSD but some revisions are needed, according to what listed below, to improve it before publication.

GENERAL COMMENTS Berx and Payne present a very interesting dataset of the North Atlantic sub-polar gyre index (SPG-1) and give a useful description of how it is computed from publically available SSH products. This index has been already used and described in several papers; nonetheless, it needs to be routinely updated - because of the effects of the lengthening time series (as shown here by the authors). Thus a major advantage could be achieved if the authors will provide regularly updates of the SPG dataset; it would be worthy to understand if the authors have any commitment/intention to do that and which is the expected timing (i.e., following SSH delayed

time products release, using near-real-time products, . . .).

Data can be easily downloaded in ASCII format and a preview tool is included in the main page to facilitate the users. No doubt that this freely available dataset could limit mistakes and uncertainties for those not used to manage altimeter data and/or principal component and empirical orthogonal analyses.

The short manuscript that comes with the index estimations is carefully written and also provides i) a useful analysis of their sensitivity to the spatial extent of the area of computation and to the length of the considered time series, ii) a comparison with index values derived in previous studies. Finally, seven appropriate figures support these analyses.

SPECIFIC COMMENTS In the abstract, I am wondering why the authors say that the sensitivity to timeseries length is not an important factor; actually it is (to me) even though it does not affect this dataset. The authors discuss this aspect later in the manuscript (section 3.2) but again it is not completely clear if timeseries length represent an issue to be carefully considered or not. This aspect should be clarified and, eventually, the sentence in the abstract should be rephrased.

In the Introduction, it would be worthy to include information about similarities and differences with the NAO index and, possibly, their combined use. Still, although the fact that this index version is better than the previous ones is clearly highlighted, the manuscript lacks an explicit description of what this index can be used for. Suggestions about how fisheries could apply this SPG index could benefit the readers and improve the use of this dataset in future studies.

In Section 3, it would be interesting to know something more about the second/third modes (how much variance they explain, what they could represent) and/or why we can neglect them when studying SPG. Generally, I would also appreciate more details about the first mode of the EOF and its physical interpretation in the SPG context.

In Figure 7, I don't think that all data shown represent "yearly mean" values as described in the figure label. Please clarify.

---

## Author Comment (AC1) · 1 Mar 2017

We would like to thank the referees for their feedback. As there was some overlap between their comments, we have addressed them in a single response. A zip file containing our responses to the three referees and a revised manuscript has been uploaded as a supplement.

Please also note the supplement to this comment:
http://www.earth-syst-sci-data-discuss.net/essd-2016-53/essd-2016-53-AC1-supplement.zip